# Ensemble and single-molecule dynamics of IFT dynein in *Caenorhabditis elegans* cilia

Jona Mijalkovic[1], Bram Prevo[1], Felix Oswald[1], Pierre Mangeol[1,†] & Erwin J.G. Peterman[1]

Cytoplasmic dyneins drive microtubule-based, minus-end directed transport in eukaryotic cells. Whereas cytoplasmic dynein 1 has been widely studied, IFT dynein has received far less attention. Here, we use fluorescence microscopy of labelled motors in living *Caenorhabditis elegans* to investigate IFT-dynein motility at the ensemble and single-molecule level. We find that while the kinesin composition of motor ensembles varies along the track, the amount of dynein remains relatively constant. Remarkably, this does not result in directionality changes of cargo along the track, as has been reported for other opposite-polarity, tug-of-war motility systems. At the single-molecule level, IFT-dynein trajectories reveal unexpected dynamics, including diffusion at the base, and pausing and directional switches along the cilium. Stochastic simulations show that the ensemble IFT-dynein distribution depends upon the probability of single-motor directional switches. Our results provide quantitative insight into IFT-dynein dynamics *in vivo*, shedding light on the complex functioning of dynein motors in general.

[1] Department of Physics and LaserLaB Amsterdam, Vrije Universiteit Amsterdam, De Boelelaan 1081, Amsterdam 1081 HV, The Netherlands. † Present address: Developmental Biology Institute of Marseille, Aix-Marseille University, 13288 Marseille Cedex 9, France. Correspondence and requests for materials should be addressed to E.J.G.P. (email: e.j.g.peterman@vu.nl).

Microtubule-based intracellular transport of organelles, receptors, messenger RNA and other cargoes is crucial for the functioning of eukaryotic cells. Cytoplasmic dyneins, members of the AAA+ ATPase family, are large multi-subunit motor complexes that are the key drivers of minus-end-directed transport along microtubules[1]. Cytoplasmic dynein 1 plays a role in a wide range of cellular processes, including cell organization, mitosis and axonal transport, while cytoplasmic dynein 2 (or IFT dynein) is specifically involved in intraflagellar transport (IFT) in cilia[2,3]. Whereas structure, motility properties and regulation of cytoplasmic dynein 1 have been widely studied[4–15], far less is known about IFT dynein.

Primary cilia project from the surface of most eukaryotic cells and have important roles in sensory perception and signalling[16]. They are built and maintained by the cooperative action of IFT-dynein and kinesin-2 motors in a bidirectional transport process along the axoneme called IFT[17–19]. In *Caenorhabditis elegans* chemosensory cilia, two anterograde motors of the kinesin-2 family, heterotrimeric kinesin-II and homodimeric OSM-3, work together to transport cargo from ciliary base to tip[20]. Kinesin-II traverses the base, transition zone and proximal segment as the 'import' motor, gradually handing over transport to OSM-3, which drives transport along the rest of the cilium up to the tip[21]. IFT dynein is the sole retrograde motor mediating the transport of turnover products from the tip back to the base[22–25].

Previous studies of IFT dynein have primarily focused on motor structure and subunit composition[23–29]. More recently, the role of different IFT-dynein subunits has been investigated in *C. elegans*[30] and *Chlamydomonas reinhardtii*[31]. Although IFT-dynein velocities and frequencies have been reported in different organisms at the ensemble level[26,28,30–34], we are not aware of studies that have addressed IFT-dynein dynamics at the single-molecule level. As a consequence, IFT-dynein motility properties remain relatively poorly characterized.

Here, we investigate IFT-dynein motility parameters at the ensemble and single-molecule level in living *C. elegans* using sensitive fluorescence microscopy. At the ensemble level, we find that directionality of IFT trains is not modulated by the kinesin-2/dynein ratio, contrary to other tug-of-war, opposite-polarity motility systems. Single-motor trajectories reveal unexpected, distinct features of IFT-dynein motility: diffusive behaviour at the ciliary base, pauses, turns, directed motion and switches between these behaviours. Stochastic simulations allow linking this single-molecule behaviour to the ensemble IFT-dynein distribution along the cilium, revealing that this distribution crucially depends upon the probability of single-motor directional switches.

## Results

**IFT-dynein ensemble dynamics.** Intracellular transport is often driven by groups of motors working together[35,36]. As such, the exact composition and size of the motor ensembles might affect motility properties and dynamics. To image *in vivo* IFT-dynein motor dynamics, we generated *C. elegans* strains with enhanced green fluorescent protein (EGFP) attached to the IFT-dynein light intermediate chain XBX-1 (ref. 28). Disruption of XBX-1 results in defective retrograde IFT in *C. elegans* cilia[28]. XBX-1 orthologues of *C. reinhardtii* (light intermediate chain, LIC) and mammals (dynein 2 light intermediate chain, D2LIC) were found to be associated with the motor heavy chain in co-immunoprecipitation assays, suggesting that XBX-1 is an integral subunit of the IFT-dynein complex[37,38]. We generated two *C. elegans* strains: one expressing genome-integrated, EGFP-tagged XBX-1 using Mos1-mediated single-copy insertion (MosSCI)[39] and another with XBX-1::EGFP expressed from an extrachromosomal array (Fig. 1a, Supplementary Tables 1 and 2

and Supplementary Fig. 1a). We first determined the steady-state motor distribution from (time-averaged) fluorescence image sequences recorded with epifluorescence microscopy. In both strains, XBX-1 is highly enriched in the base and the beginning of the transition zone, with relatively constant concentration in the rest of cilium (Fig. 1a and Supplementary Fig. 1b and 1c). In the extrachromosomal array strain, however, about twice as much XBX-1::EGFP is localized at the base and much more in the dendrite (20–30 times), guiding our choice to use the MosSCI strain for our quantitative imaging approach.

To characterize the properties of IFT-dynein ensembles moving together in trains, we recorded fluorescence image sequences and generated kymographs using *KymographClear* that uses Fourier filtering to separate anterograde and retrograde moving components (Fig. 1a)[40]. Trajectories were extracted from kymographs using *KymographDirect*[40] and further analysed to obtain position-dependent velocities (Fig. 1b,c), motor numbers (Fig. 1d), frequency (Fig. 1e,f) and flux (Fig. 1g). IFT dynein-driven trains exhibit relatively constant velocity of $\sim 1.7\,\mu m s^{-1}$ along distal and proximal segments of the *C. elegans* cilium, similar to previously reported retrograde IFT velocities[30,34,41]. In the transition zone, however, we observe the slowing down of IFT-dynein trains. A similar deceleration has been observed before for IFT particle B, OSM-3 and kinesin-II and is most likely due to transition zone structures impeding transport[21]. IFT-dynein motor numbers were calculated from the fluorescence intensities of trains as described before[21], assuming that each IFT-dynein dimer consists of two XBX-1::EGFP subunits[42]. We find that the number of IFT dyneins per train remains relatively constant during transport from transition zone to the tip and vice versa (Fig. 1d). However, whereas anterograde trains are composed of 40–50 IFT-dynein complexes as cargo, retrograde trains contain only 20–30 motors driving motion. In a previous study, we observed a proportionally comparable decrease of kinesin motors and IFT-particle components on retrograde trains compared with anterograde trains[21]. Electron microscopy studies of *C. reinhardtii* have shown a similar decrease in train length[43]. In more recent studies, a difference in size between retrograde and anterograde trains was not observed, but retrograde trains appeared morphologically different compared with anterograde trains[44,45]. Automated detection of peaks in vertical kymograph intensity profiles (that is, intensity at a given position as a function of time) (Fig. 1e,f) shows that retrograde trains are $\sim 1.6$ times more frequent than the anterograde trains. Similarly, in *Trypanosoma brucei* and *C. reinhardtii* cilia, retrograde frequency was observed to be higher[31,32].

To obtain further quantitative insight into IFT-dynein trafficking, we calculated XBX-1 flux in the cilia. Flux is defined here as the number of motors (calculated from the area under fluorescence intensity versus time profiles such as Fig. 1e,f) multiplied by the local velocity (Fig. 1g). In this way, all IFT dyneins are taken into account, including small trains that might not be discernible in kymographs[32]. We find that retrograde and anterograde XBX-1 fluxes are very similar (Fig. 1g), implying that the amount of IFT dynein motors in the cilium is constant in time and that there is no substantial loss of IFT dynein at the ciliary tip. Taken together, our results indicate that retrograde trains are more frequent but smaller than anterograde trains, resulting in equal anterograde and retrograde fluxes.

**Directionality is not modulated by kinesin-2/dynein ratio.** We next addressed how IFT dynein cooperates with the antero-grade kinesin-2 motors. Different modes of opposite-polarity motor cooperation have been proposed in literature, including mechanical-competition models (tug-of-war) and coordinated,

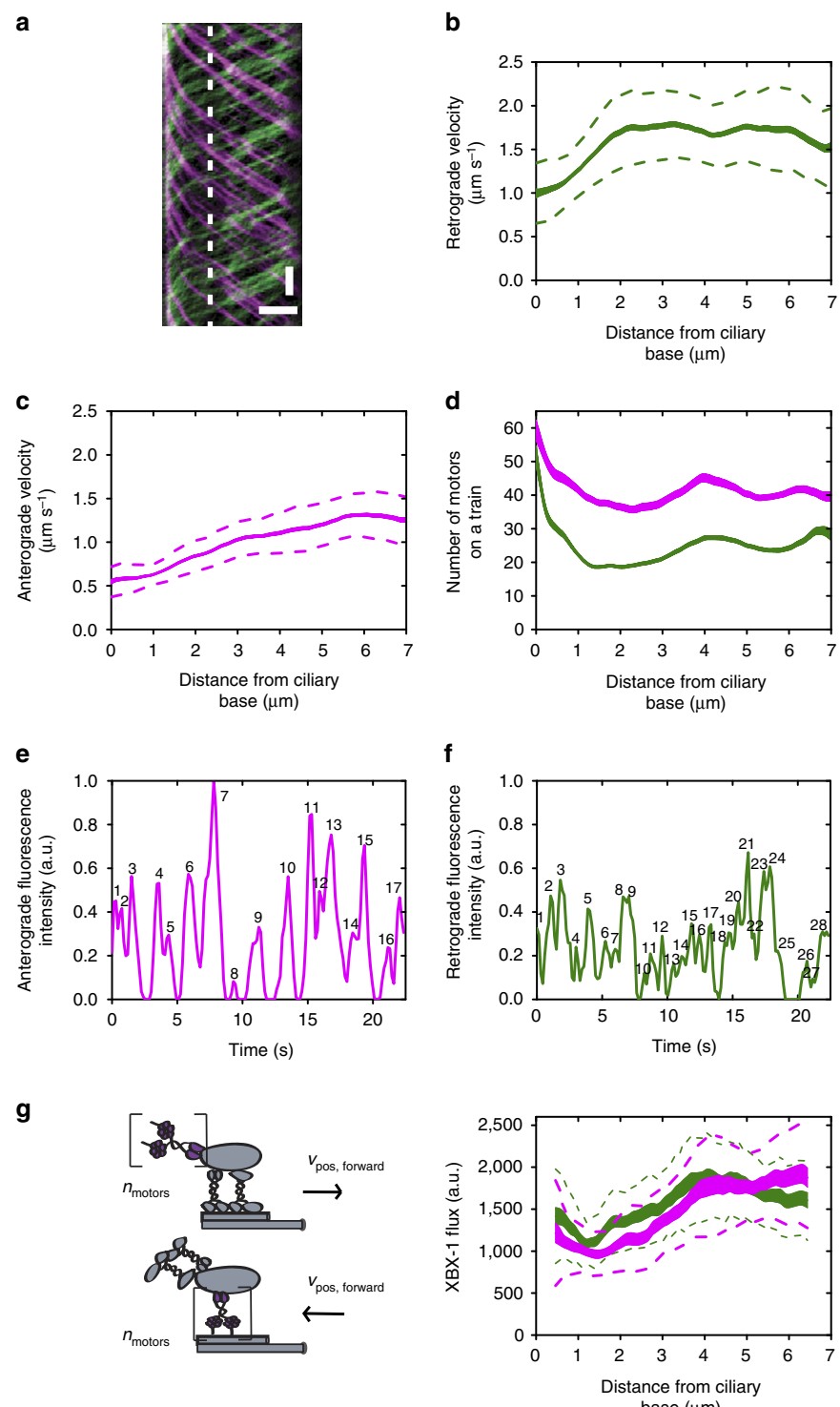

**Figure 1 | Characterization of IFT-dynein ensemble dynamics.** (**a**) Representative XBX-1::EGFP kymograph showing retrograde (green) and anterograde (magenta) motility. Time: vertical; scale bar, 2 s. Position: horizontal; scale bar, 2 μm. (**b**) Train-averaged retrograde IFT-dynein velocity profile. (**c**) Train-averaged anterograde IFT-dynein velocity profile. (**d**) Train-averaged retrograde (green) and anterograde (magenta) XBX-1 train intensity. (**e,f**) Representative intensity versus time profiles of anterograde (**e**) and retrograde (**f**) motility components in (**a**) (location indicated by dotted vertical line). Black numbers are peaks identified using automated peak detection. (**g**) Motor flux is calculated by multiplying the number of motors (area under the curve of the position-dependent fluorescence intensity over time) by the position-dependent velocity. Shown is XBX-1 retrograde (green) and anterograde (magenta) flux in the cilium, $n = 30$ worms, (**b–d**) averaged over 529 trains obtained from 30 worms (retrograde) and 425 trains from 30 worms (anterograde). Line thickness represents s.e.m.; dotted line represents s.d. See also Supplementary Fig. 1.

regulated action models[46,47]. To gain more insight into opposite-polarity motor cooperation in *C. elegans* chemosensory cilia, we generated double-labelled MosSCI strains XBX-1::EGFP OSM-3::mCherry and XBX-1::EGFP KAP-1::mCherry (KAP-1 is a subunit of kinesin-II; Supplementary Table 1). We imaged the strains using two-colour fluorescence microscopy to

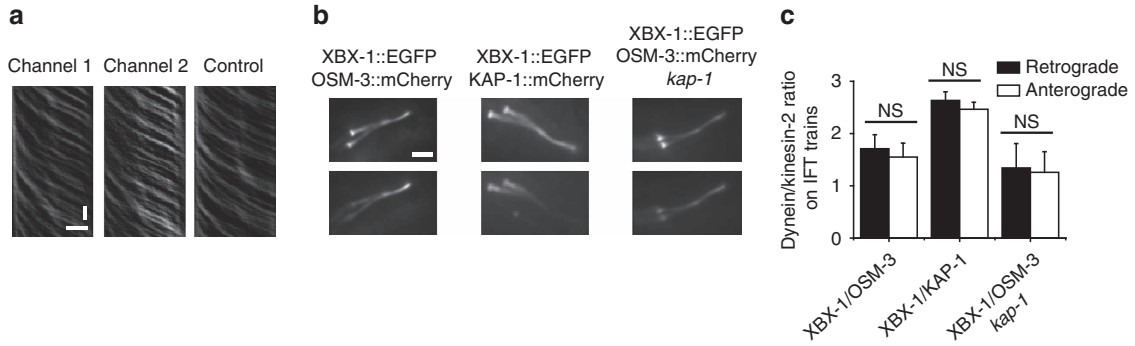

**Figure 2 | Directionality is not modulated by the dynein/kinesin-2 ratio. (a)** Representative Fourier-filtered anterograde kymographs of XBX-1::EGFP (channel 1, M1), OSM-3::mCherry (channel 2, M2) and a randomly chosen control kymograph used for Manders' colocalization analysis. Time: vertical; scale bar, 2 s. Position: horizontal; scale bar, 2 μm. **(b)** Time-averaged fluorescence images of dual-colour strains XBX-1::EGFP OSM-3::mCherry, XBX-1::EGFP KAP-1::mCherry and XBX-1::EGFP OSM-3::mCherry *kap-1*. Scale bar, 2 μm. **(c)** The dynein/kinesin-2 ratio on trains was generated using Fourier-filtered kymographs from dual-colour constructs for each direction of movement (see also Supplementary Fig. 2). A line is drawn along the same train of interest in both channels and corrected for background, photobleaching, bleed-through and excitation intensity. Division of the two corrected position-dependent intensities gives the XBX-1/OSM-3 ratio ($n = 95$ trains, 24 worms), XBX-1/KAP-1 ratio ($n = 67$ trains from 20 worms) and XBX-1/OSM-3 *kap-1* ratio ($n = 56$ trains from 14 worms). The XBX-1/OSM-3 ratio was taken in the distal segment region (4.0–7.5 μm) and the XBX-1/KAP-1 ratio in the proximal segment region (0–3.5 μm). Error is s.d. NS, no significant difference, unpaired *t*-test. See also Supplementary Table 3 and Supplementary Figs 2 and 3.

simultaneously visualize IFT dynein and either kinesin-2 motor (Fig. 2a and Supplementary Movie 1). Visual inspection of kymographs alluded that all OSM-3- or kinesin-II-containing trains also carry IFT dynein (Supplementary Fig. 2). To further substantiate this observation, we registered the two colour channels on the camera and determined the Manders' colocalization coefficients for both directions of motion from the Fourier-filtered, background, bleaching and bleed-through corrected kymographs (Fig. 2a). The Manders' coefficients M1 and M2 are measures of how well fluorescence intensity of pixels in the first channel colocalize with those in the second channel, and vice versa[48]. To account for the ciliary locations of KAP-1 and OSM-3, we focused, for each kinesin motor, on a specific ciliary segment: the distal region (4.0–7.5 μm) for XBX-1::EGFP OSM-3::mCherry and the proximal region (0–3.5 μm) for XBX-1::EGFP KAP-1::mCherry (Fig. 2b). For XBX-1 and OSM-3, both Manders' coefficients are close to one (M1 = 0.92 ± 0.01; M2 = 0.90 ± 0.01, mean ± s.e.m., $n = 24$) and equally so for XBX-1 and kinesin-II (M1 = 0.92 ± 0.01; M2 = 0.82 ± 0.02, $n = 18$). These values are significantly larger than control correlations of two different, same-directionality OSM-3 (M1 = 0.40 ± 0.03; M2 = 0.44 ± 0.02) or kinesin-II kymographs (M1 = 0.43 ± 0.03; M2 = 0.45 ± 0.03, Supplementary Table 3). This analysis shows that most, if not all, IFT trains contain both anterograde and retrograde motors. In addition, whereas kinesin-II is proposed to undergo diffusion in *C. reinhardtii* flagella[22], our results show that in *C. elegans* kinesin-II is actively carried back to base by IFT dynein, in agreement with previous findings[21].

For tug-of-war motor systems, it has been shown that the transport direction is modulated by the ratio of opposite polarity motors[49,50]. To test whether such a mechanism might be at play in *C. elegans* IFT, we determined the ratio of IFT dynein and kinesin-2 motors on anterograde and retrograde IFT trains (Supplementary Fig. 2) by evaluating the ratio of the fluorescence intensities of the two motors (Fig. 2c). On average, the XBX-1/KAP-1 fluorescence-intensity ratio is 2.47 ± 0.13 on anterograde (mean ± s.d. for 67 trains in 20 animals; averaged from 0 to 3.5 μm along the cilium) and 2.63 ± 0.16 on retrograde trains (61 trains in 20 animals). The XBX-1/OSM-3 ratio is 1.55 ± 0.27 on anterograde (95 trains in 24 animals; averaged

from 4 to 7.5 μm along the cilium) and 1.71 ± 0.27 on retrograde trains (86 trains in 23 animals). These results reveal that there is no significant difference in dynein/kinesin-2 ratio between retrograde and anterograde trains, which strongly suggests that, in *C. elegans* IFT, directionality is not modulated by the ratio between opposite-polarity motors.

To further probe the interplay between IFT dynein and the kinesin-2 motors, we generated a double-labelled XBX-1::EGFP and OSM-3::mCherry strain that lacks functional kinesin-II (Supplementary Table 1 and Supplementary Movie 1). Steady-state intensity profiles show that the overall number of XBX-1 and OSM-3 in the cilium is decreased (Supplementary Fig. 3), in agreement with previous experiments on OSM-3 (ref. 21). XBX-1 and OSM-3 motor numbers are also lower on anterograde and retrograde trains (Supplementary Fig. 3). Intriguingly, the IFT dynein/OSM-3 ratios are similar to those in the constructs with functional kinesin-II (Fig. 2c). This observation suggests that IFT dynein/kinesin-2 ratios are tightly controlled in the IFT system, for example by the limited and fixed number of motor binding sites on the IFT particle complex[51].

**Single-molecule quantification.** To understand how the dynamics of individual IFT-dynein motors result in the overall ensemble behaviour discussed above, we investigated the XBX-1 motility properties at the single-molecule level. To access the individual dynein regime, we generated *C. elegans* strains expressing XBX-1::paGFP (Supplementary Table 1). After photoactivation of only a limited number of paGFPs, we could visualize individual XBX-1 in living *C. elegans* for the first time (to our knowledge). Analysis of the recorded image stacks using single-particle tracking yielded 494 single-XBX-1 trajectories from 7 worms. These trajectories were located along the entire cilium, except for the last 1–2 μm close to the ciliary tip. This part of the phasmid is commonly bent with respect to the rest of the cilium and was, as a consequence, out of focus. Careful inspection of the trajectories revealed different single-motor behaviours (Supplementary Movie 2).

Most of the single motors moved unidirectionally in either anterograde (182 trajectories; 37% of all 494) or retrograde

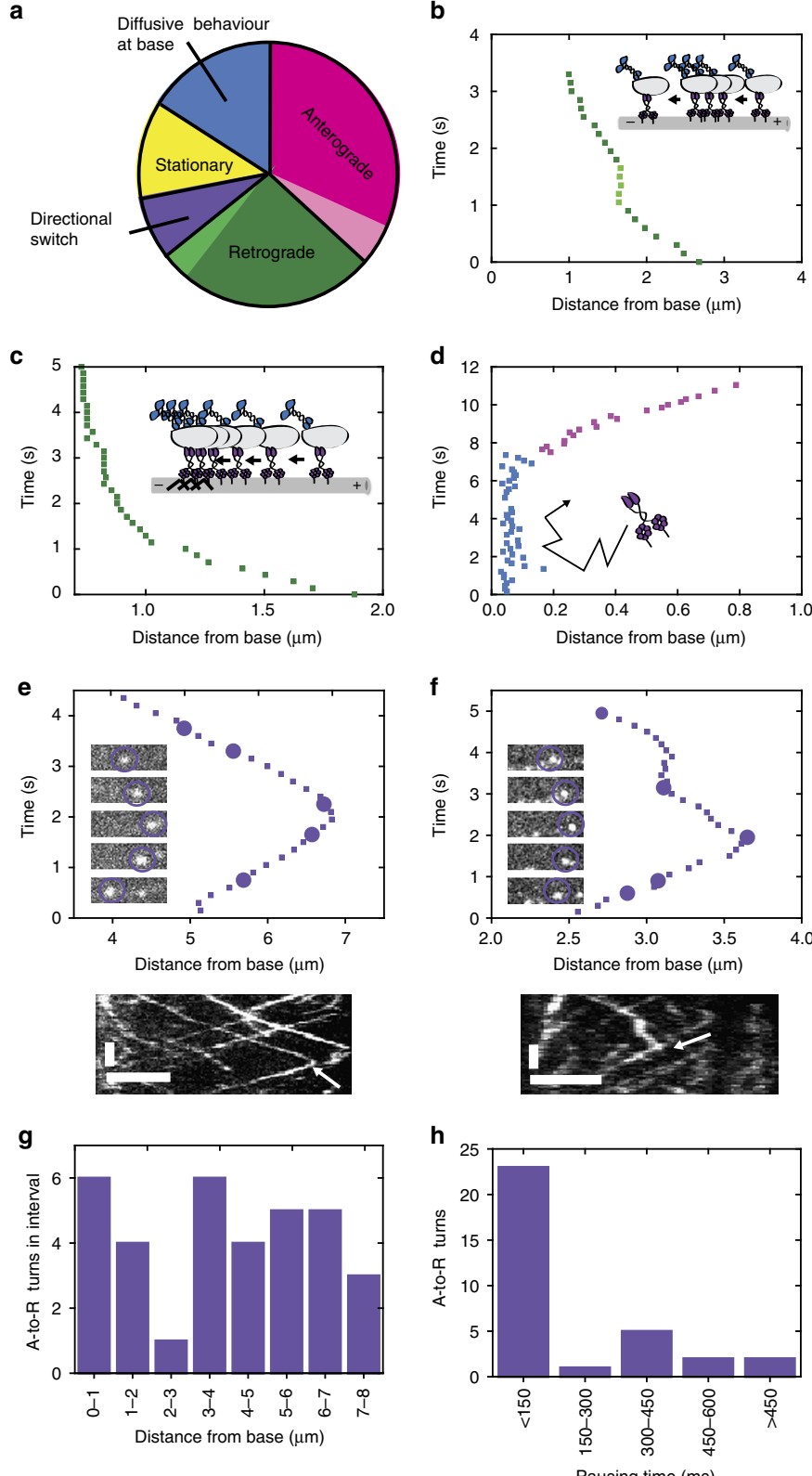

**Figure 3 | Single-molecule quantification reveals novel IFT-dynein behaviour. (a)** Quantification of single-motor IFT-dynein trajectories: 37% anterograde motion (magenta), of which 14% with pausing (light magenta); 27% retrograde motion (green), of which 13% with pausing (light green); 16% at the base (blue); 12% stationary (yellow); 8% turnarounds (purple); $n = 494$ trajectories from 7 worms. Only trajectories of 600 ms or longer are taken into account. **(b–f)** Representative single-motor IFT-dynein traces showing: pausing (light green) in unidirectional trajectories **(b)**, deceleration upon reaching the ciliary base **(c)**, diffusive behaviour at the ciliary base **(d)** and A-to-R (anterograde to retrograde) directional switches at different positions with corresponding images and kymograph **(e,f)**. **(e,f)** Time: vertical; scale bar, 2 s. Position: horizontal; scale bar, 2 μm. Arrow indicates turnaround position. The movie corresponding to **(f)** is shown in Supplementary Movie 2. **(g,h)** Histograms of IFT-dynein A-to-R turnarounds ($n = 34$) showing their location and pausing time. R-to-A not shown (only four recorded trajectories).

(135 trajectories; 27%) direction (Fig. 3a). A smaller fraction of trajectories showed diffusive behaviour (81 trajectories; 16%; Fig. 3d). These trajectories were primarily located at the ciliary base, congruent with XBX-1 enrichment in this region (Supplementary Fig. 1b). Only a small fraction of the trajectories (58; 12%) corresponded to stationary motors. Further analysis of the retrograde single-molecule trajectories confirmed that IFT dyneins slow down upon approaching the ciliary base, as observed at the ensemble level (Fig. 3c). Collectively, these findings are in agreement with our data on the ensemble motility properties of XBX-1, where kymographs show continuous train movement from base to tip, driven by kinesins, with XBX-1 as cargo and from tip to base, driven by IFT dynein (Fig. 1a).

We then turned our focus to the characteristics of individual IFT dynein motility that cannot be observed at the ensemble level. First, we noticed that both anterograde and retrograde trajectories were sometimes interrupted by pauses (anterograde: 26 trajectories, 14% of the 182 anterograde trajectories; retrograde: 18 trajectories, 13% of the 135 retrograde trajectories; Fig. 3b). The average duration of these pauses was $0.71 \pm 0.20$ s (s.d.), with the longest pause lasting 1 s. Second and unexpectedly, we observed directional switches of IFT dynein (38 trajectories, 8%). The vast majority (34; 89% of 38 directional switches) were in the anterograde-to-retrograde direction (A-to-R; Fig. 3e,f and Supplementary Movie 2). In these trajectories, IFT dynein appears to switch from being a cargo on a kinesin-driven anterograde IFT train to an active transporter moving in the retrograde direction. A histogram of the locations where A-to-R turnarounds took place indicates that there is no A-to-R turnaround hotspot, but rather these directional switches can occur at all locations in the cilium (Fig. 3g). Furthermore, a histogram of pause durations during A-to-R switches (Fig. 3h) reveals that the majority of switches (23; 64%) is faster than 150 ms (the time between subsequent frames in our measurements). To obtain further insight into directional switches, we determined the probability density (probability per µm travelled) of an IFT dynein to switch from A-to-R ($P_{AR}$) or R-to-A ($P_{RA}$) by calculating the total number of A-to-R (or R-to-A) turnarounds divided by the total anterograde (or retrograde) distance travelled. We found a $P_{AR}$ of $0.14 \pm 0.03 \, \mu m^{-1}$ and $P_{RA}$ of $0.07 \pm 0.03 \, \mu m^{-1}$; that is, the probability density of anterograde IFT dyneins reversing is twice as high as retrograde IFT dyneins. As a consequence, an IFT dynein motor traveling from the base in the anterograde direction has a probability of $28 \pm 7\%$ of reaching the ciliary tip (assuming 9 µm ciliary length). For a retrograde-moving IFT dynein the probability to reach the base when it started at the tip is $53 \pm 12\%$. This difference in turnaround probability between IFT dynein moving in anterograde and retrograde direction highlights an important difference between anterograde and retrograde trains: IFT dynein appears more strongly coupled to trains when it is active as driver than when it is passive as cargo.

**Stochastic modelling**. To connect the observed single-molecule behaviour to the overall IFT-dynein distribution over the cilia, we performed stochastic simulations. In these simulations, we allowed an IFT-dynein motor to take nanometre steps along a 9 µm cilium and stochastically reverse direction. As input parameters we used the experimentally determined, position-dependent IFT dynein train velocities and the turnaround probability densities $P_{AR}$ and $P_{RA}$. We assumed that a motor turned around upon reaching the ciliary tip (without pause) or base (with an exponentially distributed pause duration $t_{RA \, base}$ of on average 500 ms, Fig. 4a). The simulations yielded time-averaged motor distributions in the cilium and histograms of the turnaround locations as output (Fig. 4a,b). The experimental histograms and

distributions can be reproduced well using this simple model. The simulated turnaround histograms indicate that compliant with our experimental findings, A-to-R and R-to-A turns occur all along the cilium (Fig. 4b). However, they also indicate that most turns must occur at tip and base, which we did not observe in these single-molecule trajectories. For turns at the tip, this can be explained by the spatial orientation of the tip that is not in the same focal plane as the rest of the cilium, therefore making it impossible to detect single molecules around the tip in these experiments. This out-of-focus effect is less of a problem in our ensemble data that indicate efficient and fast turnarounds of XBX-1 at the tip (Fig. 1a), as observed before[30]. To obtain deeper insight into turnaround behaviour of XBX-1 specifically at the tip, we recorded single-molecule trajectories in a small subfraction of C. elegans that were oriented such that the distal segment including tip was located in one image plane (Supplementary Fig. 4). From these trajectories we determined the XBX-1 pause duration during tip turnaround and found that the majority of switches were almost instantaneous ( < 600 ms; 67% of 54 trajectories). At the base, we attribute the failure of experimentally detecting all turns to the pausing and diffusing of motors at this location that substantially reduces the chance of detecting turns in the relatively short trajectories, whose duration is limited by photobleaching. Further simulations with altered turnaround probability densities show that motor distribution is highly sensitive to tuning the pausing time and turnaround probability, with relatively small modifications of the probabilities resulting in substantially altered motor distributions (Fig. 4c and Supplementary Fig. 5). This suggests that the interaction between IFT dynein and the IFT train is an important parameter regulating IFT.

**Discussion**
IFT dynein is the retrograde motor that drives IFT in cilia and flagella and is essential for ciliary assembly and signalling. Here, we probed its ensemble and single-molecule dynamics in the chemosensory cilia of C. elegans, providing the first single-molecule characterization of IFT-dynein motility in vivo. A key single-molecule observation was that some individual IFT-dynein motors switch direction along the entire length of the cilium. We determined that the turnaround probability densities are $0.14 \pm 0.03 \, \mu m^{-1}$ for A-to-R turns and $0.07 \pm 0.03 \, \mu m^{-1}$ for R-to-A turns, corresponding to effective average anterograde and retrograde run lengths of $7 \pm 1$ and $14 \pm 1 \, \mu m$, respectively. It is important to note that these in vivo run lengths indicate the distance along which motors remain attached to a train either as cargo or as active driver of transport. We can only observe that XBX-1 is attached to a moving IFT train. On anterograde trains, kinesin is the active motor; IFT dynein is cargo and thus not active as motor. On retrograde trains, at any given time at least a fraction of the IFT-dynein motors must be involved in driving the transport. From in vitro motor stall force measurements, it has been estimated that at least four motors actively drive transport at a given time in C. reinhardtii[34], but the rest might effectively be cargo. Here we cannot distinguish between these two roles. As such, these values do not correspond to run lengths measured in vitro on isolated motors using optical tweezers or single-molecule fluorescence microscopy that have not been determined for IFT dynein, but are typically 1–3 µm for cytoplasmic dynein 1 (refs 6,52,53). Furthermore, these results indicate that IFT dynein is more likely to reverse direction as passenger than as driver. This suggests that the interaction between IFT dynein and the IFT-particle backbone is stronger when IFT dynein drives retrograde transport than when it is carried as anterograde cargo. This could be due to direction-dependent differences in train

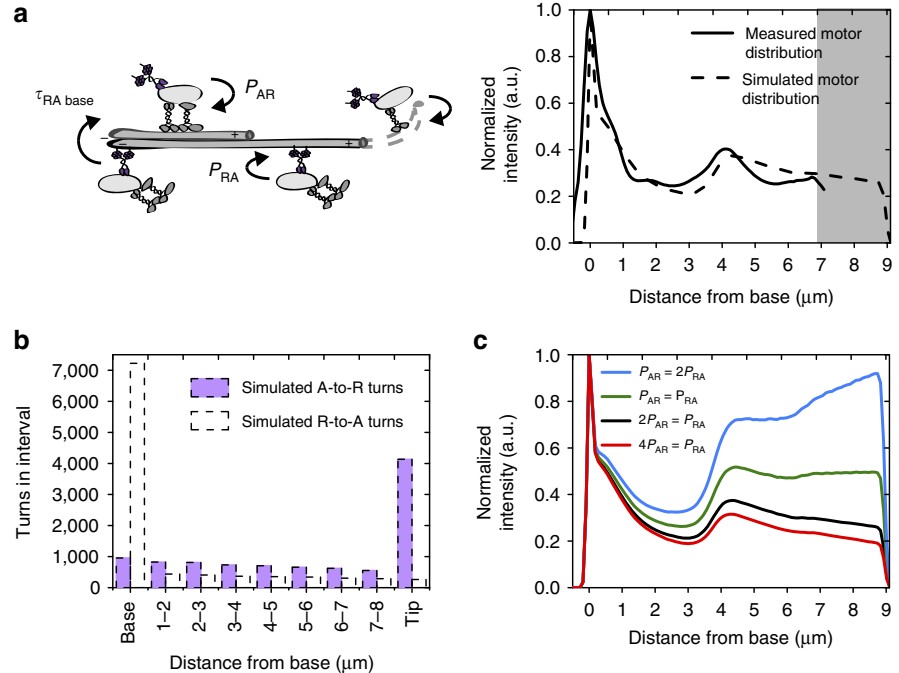

**Figure 4 | Stochastic modelling connects dynamics of individual IFT dyneins to ensemble behaviour.** (**a**, left) Cartoon highlighting stochastic simulations of single IFT dyneins walking and turning around in the cilium. The input parameters of the simulations are indicated. The probability densities of an IFT dynein A-to-R turn ($P_{AR}$, 0.14 μm$^{-1}$) and an R-to-A turn ($P_{RA}$, 0.07 μm$^{-1}$), were obtained experimentally. Pausing time for A-to-R turns ($\tau_{AR}$, no pause) and R-to-A turns at the base ($\tau_{RA}$, 500 ms) were estimated. (**a**, right) Measured (black line) and simulated (dotted line) IFT dynein steady-state distribution in the cilium. In the simulations, we assumed that all IFT-dynein motors in the tip region (shown in grey; not in focus in experimental measurements) turn. (**b**) Simulated numbers and positions of A-to-R (dotted, black) and R-to-A (dotted, purple) turns. (**c**) The effect of $P_{AR}$ and $P_{RA}$ on IFT-dynein motor distribution in stochastic simulations. Motor distribution in black was obtained using probability parameters calculated from the single-molecule experiments ($P_{AR} = 0.14$ μm$^{-1}$, $P_{RA} = 0.07$ μm$^{-1}$) and most closely corresponds to the IFT-dynein distribution from ensemble measurements (same as solid line in (**a**)). Green ($P_{AR} = 0.14$ μm$^{-1}$, $P_{RA} = 0.14$ μm$^{-1}$), blue ($P_{AR} = 0.07$ μm$^{-1}$, $P_{RA} = 0.14$ μm$^{-1}$) and red ($P_{AR} = 0.28$ μm$^{-1}$, $P_{RA} = 0.07$ μm$^{-1}$) are simulated distributions of altered turnaround probabilities. To account for the overlap of two phasmid cilia in the distal segment, intensities and histograms were multiplied by $2 - (1/(e^{(x - 3,500)/200} + 1))$. See also Supplementary Fig. 5.

properties or tension-dependent interactions between IFT dynein and particle. Another explanation could be that active motors remain bound to the microtubule after detachment from a train that could allow rebinding to the train.

How do these single-molecule IFT-dynein turnarounds relate to our ensemble findings that trains of IFT dynein appear to switch direction only at base and tip of the cilium? To shed more light on this apparent inconsistency, we performed stochastic simulations of IFT-dynein runs. The simulations confirm that although most motors turn at the base and tip, turnarounds along the entire cilium are necessary to maintain the IFT-dynein distribution: a $P_{AR}$ that is twice of $P_{RA}$ results in IFT-dynein distribution that corresponds closely to the experimentally observed one, highlighting the significance of direction-specific motor and/or train properties as discussed above. Contrary to IFT dynein, the IFT-B particle subcomplex only switches direction at the base and tip and does not turn around along the entire cilium[21]. An overall picture that emerges for IFT in *C. elegans* is that IFT trains consisting of multiple particles IFT-A, IFT-B and BBSome are assembled at tip and base and traverse the cilium as a whole, forming a stable scaffold[21,41]. The motors, kinesin-II (ref. 21), OSM-3 (ref. 21) and IFT dynein (this work), gradually dock on and off to this scaffold in a tightly controlled way, with conserved flux. Tight control of the binding and release of the motors ensures that the motors are distributed differently along the cilia: the importer kinesin-II enriched at base and transition zone, the long-distance transporter OSM-3 in proximal and distal

segment and the sole retrograde motor IFT dynein relatively constant along the cilium.

A recent ensemble-level study has shown that most anterograde-to-retrograde XBX-1 turns occur at the ciliary tip[30]. This is consistent with our ensemble data, single-molecule data and simulations that show almost instantaneous XBX-1 turns at the tip. In *C. reinhardtii*, however, an IFT-dynein light intermediate chain has been shown to pause substantially during tip turnaround (~1.7 s)[31]. In addition, *C. elegans* IFT-dynein components DYLT-3, DLC-1 and DYCI displayed turnaround behaviours different from XBX-1. This raises questions about whether those IFT-dynein components show similar dynamics at the single-molecule level as XBX-1, what role each component plays in retrograde transport and whether there are differences between species. Despite recent progress in identifying IFT-dynein subunits, the exact motor composition remains elusive[27]. To fully understand retrograde transport, it will be vital for future structural and functional studies to uncover which dynein-associated subunits are an integral component of the motor and which act as regulatory, accessory or adapter proteins.

Our findings not only provide insight into dynein dynamics in IFT, but are also valuable in understanding the dynamics of cytoplasmic dyneins in general. While numerous studies have reported characteristics of cytoplasmic dynein 1 motility *in vitro*, including side and back stepping[6,54,55], response to load[56] and collective behaviour[57], *in vivo* studies have been lacking. Further studies will be necessary to discern whether cytoplasmic dynein 1

exhibits similar *in vivo* dynamics to IFT dynein and, if so, what the implications are for its cellular functions.

## Methods

**C. elegans strains and generation of constructs.** *C. elegans* maintenance and genetic crosses were done using standard procedures. Nematodes were grown at 20 °C on Nematode Growth Medium plates seeded with *Escherichia coli* OP50 bacteria. The *C. elegans* strains used in this study are listed in Supplementary Table 1. Most strains were constructed using MosSCI[21,39]. The gene of interest, including regions up- and downstream to the neighbouring genes, was fused to codon-optimized fluorescent protein using Gibson assembly and verified by sequencing before injection[58,59]. *unc-119* animals with a Mos1 insertion site were injected with a phasmid mix containing the gene of interest, co-injection selection markers (prab-3::mCherry, pmyo-2::mCherry and pmyo-3::mCherry) and transposase[39]. Coordinated worms lacking the mCherry signal were selected and integration of the transgene was confirmed by PCR of regions spanning outside and inside the insertion (Supplementary Table 2). Extrachromosomal array strain EJP217 was generated using microinjection of the fluorescently labelled gene of interest into the gonad.

**Fluorescence imaging and intensity quantification.** Fluorescence images were obtained using a custom-built epi-illuminated fluorescence microscope described in detail in Prevo et al.[21]. *C. elegans* young adult hermaphrodites were anaesthetized in 5 mM levamisole in M9 and immobilized on a 2% agarose in M9 pad covered with a 22 × 22 mm cover glass and sealed with VaLaP. Then, 150 frames were recorded for each phasmid cilium at 152 ms per frame and 300 EM gain. The cilia were chosen based on their orientation plane, with the base, proximal segment and distal segment in focus (but not the ciliary tip). To quantify the number of motors on IFT trains we measured single EGFP intensities *in vitro*, as described previously[21]. To quantitatively compare EGFP and mCherry intensities, we measured the cilium-averaged integrated fluorescence intensity of KAP-1::EGFP (EJP13) and KAP-1::mCherry (EJP85) at different excitation powers. A correction factor was obtained from the slope of fluorescence intensity versus excitation power[21]. The single-paGFP localization accuracy of our instrument under the conditions used here is 40 nm (ref. 21).

**Image analysis.** Ensemble data images were analysed using kymographs generated by the open source tools KymographDirect and KymographClear developed in our lab[40]. Single-molecule images were analysed using a tracking and linking algorithm in MATLAB (Mathworks)[60]. A spline was drawn along the cilium to define the *x* axis of displacement; *y* coordinates are perpendicular to the spline. A trajectory was classified as an event when at least 600 ms long with minimum displacement of 200 nm. Particles with interframe displacement of <50 nm, consecutively over at least 4 frames, were defined as stationary. Stationary behaviour within retrograde trajectories, anterograde trajectories or directional switches was defined as pausing. A directional switch or turnaround event has a local maximum or minimum in the *x* coordinate. A minimum of three consecutive steps in the same direction are taken before and after the turn.

**Manders' colocalization coefficients.** Pixels in EGFP and mCherry channels are aligned post imaging using a custom-written MATLAB algorithm using non-moving features (for example, worm body wall) for alignment and corrected for background, bleaching and EGFP bleed-through into the mCherry channel. Thresholded Manders' coefficients M1 and M2 were generated using the Coloc2 plugin[48,61] on background, bleaching and bleed-through-corrected, aligned, Fourier-filtered kymographs from the EGFP and mCherry channels using a region of interest depending on the localization of the motor protein (transition zone and proximal segment, 0–3.5 μm, for KAP-1::mCherry; distal segment, 4.0–7.5 μm, for OSM-3::mCherry). A different, same-directionality kymograph from the same data set was used as a control. To minimize the effects of photobleaching, only the first 100 frames of each movie were used.

**Stochastic simulations of IFT-dynein dynamics.** Stochastic simulations were performed using a custom-written algorithm in LABVIEW (National Instruments). Single motors moved repetitively along a 9,000 nm track with 1 nm steps and experimentally measured velocity (10,000 events). Motors turned around stochastically based experimentally determined probability of turnaround over distance, $P_{AR}$ (anterograde to retrograde) and $P_{RA}$ (retrograde to anterograde). To account for the overlap of two phasmid cilia in the distal segment, intensities and histograms were multiplied by $2 - (1/(e^{(x - 3,500)/200} + 1))$. Results were convoluted with a Gaussian function of width identical to the point-spread function of the microscope to account for its limited spatial resolution. The simulation software can be downloaded at: http://www.nat.vu.nl/~erwinp/downloads.html

**Data availability.** The authors declare that the data supporting the findings of this study are available within the article (and its Supplementary Information Files) or available from the authors on request.

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

## Acknowledgements

We thank J. Girard and D. Vorselen for help with data analysis, J. van Krugten for his help with the single XBX-1 tip-turn experiments, M. Bos for cloning the XBX-1::paGFP construct, S. Acar and J.M. Scholey for helpful discussions and advice and E. Kroezinga for biochemical support. Several strains were provided by the CGC that is funded by NIH Office of Research Infrastructure Programs (P40 OD010440). We acknowledge financial support from the Netherlands Organization for Scientific Research (NWO) via a VICI grant, an NWO-Groot and an ALW Open Program grant, and from NanoNextNL of the Government of the Netherlands and 130 partners.

## Author contributions

Conceptualization: J.M., B.P. and E.J.G.P.; methodology: J.M., B.P., P.M., F.O. and E.J.G.P.; investigation and formal analysis: J.M.; software: F.O. and P.M.; resources: J.M. and B.P.; writing, original draft: J.M. and E.J.G.P.; writing, —review editing: J.M., B.P., P.M., F.O. and E.J.G.P.; funding acquisition and supervision: E.J.G.P.

## Additional information

**Competing financial interests:** The authors declare no competing financial interests.

