## [Peer Review File · Nature Communications]

Reviewers' comments:

Reviewer #1 (Remarks to the Author):

"Backward, forward, pausing, turning: single-molecule behaviour required to maintain IFT dynein distribution in *C. elegans* chemosensory cilia" by Jona Mijalkovic, Bram Prevo, Felix Oswald, Pierre Mangeol, and Erwin J.G. Peterman

This study uses state-of-the-art in vivo imaging to study intraflagellar transport in *C. elegans*, in particular the behavior of fluorescent protein-tagged XBX-1, a light-intermediate chain of the retrograde IFT motor IFT dynein. IFT dynein is studied at the ensemble level (= IFT train) and the single molecule level. The manuscript shows high quality imaging (imaging single dynein motors in whole animals !!), is clearly written, and the data are solid.

The manuscript addresses two main questions: First, considering that IFT trains are associated with both kinesins and dyneins, how is a tug-of-war avoided and processive transport of the ensembles over a distance of several microns ensured? The authors show that the ratio of anterograde and retrograde motors on the IFT trains is essentially constant. This is novel and interesting, but fits what was widely assumed and barely touches the question of how a tug-of-war between the opposing motors is avoided.

The second question concerns the behavior of individual dynein motors and how it relates to and leads to the behavior of the ensembles (trains). Using modeling it is shown that the surprising variation of single molecule behavior (including diffusion, stationary, and turn-around) can explain the overall distribution of dynein. I have several questions about that approach and definiteness of the outcome (see below).

I feel that the current paper, while excellently executed and revealing various novel aspects of IFT, which will be of high interest to the cilia community, does not provide enough progress for publication in the journal.

Major points

Simulations were used to connect single molecule behavior to the overall distribution of IFT. I have several questions about this approach.

1. I don't understand how the probabilities for anterograde-to-retrograde (pAR) and retrograde-antegrade (pRA) turns were obtained. Out of the 38 turnarounds observed just 4 were of the RA type. Retrograde IFT trains are 1.6 times more frequent than anterograde ones and that both AR and RA trains travel the same distance. The AR probability was given as 0.14/ μm and pRA as 0.07/ μm . Considering that RA turnarounds are apparently much less frequent than AR turns (factor 8.5), how do the ps end up so similar (factor 2)?

2. The simulation assumes that dynein at the cilia tip turns-around without a pause. I wonder if that is a reasonable assumption considering data in other species documenting a pause of several seconds for IFT proteins at the cilia tip (see recent analysis by Reck et al. on the light intermediate chain D1bLIC-GFP and Buisson et al. in *Trypanosoma*; Reck et al. also determined A:R ratios in IFT frequency etc. and should be cited and discussed.)

3. The conclusions are based on the analysis of a single marker (XBX-1). Playing devil's advocate, I wonder to which extent XBX-1's behavior is representative for the entire IFT dynein complexes? Or that the GFP-tag results in an aberrant behavior of the protein/protein complex? It should be mentioned somewhere that this light chain is stably integrated into the dynein complex (co-purifies?? reference?). More difficult to exclude is the possibility that entire pieces of an IFT train (dynein + IFTA and/or IFT-B) brake-off and re-attach. Then, the observed single molecule features would be those of the IFT train subunits rather than of the dynein motor. This could actually be tested experimentally using an IFT particle proteins and/or a second IFT dynein subunit. Or do the authors dismiss this possibility?

Minor points

The terms 'backward', 'switch direction' and similar could be easily misinterpreted as dynein actually taking a step as a motor toward the microtubule plus-end as it has been reported for dynein-1. Actually, XBX-1 switches from being a cargo (A) to being an active (or inactive) motor (R).

The axes in the plot in Fig. 3b should be oriented as in the other subpanels (c-f). Also, it would be nice to show kymographs in addition to the particle tracings.

p.4 length of A and R trains: reference 28 has been superseded by Vannuccini et al. 2016 and especially Stepanek and Gaia 2016.

p. 12, "...roles we cannot distinguish between." However, in several other places in the manuscript (e.g., line 308/309 on the same page) the authors do distinguish and assume that IFT dynein is a cargo on anterograde trains (reasonable) but active in retrograde trains (probable, but who knows how many of the dyneins in such an ensemble are actively moving and how many are carried along as cargo?

Reviewer #2 (Remarks to the Author):

The manuscript by Mijalkovic et al. reports on the successful (genetic) fluorescence tagging of cytoplasmic dynein 2 (IFT dynein) in *C. elegans* strains and the imaging of the tagged dyneins in vivo both at the ensemble and single-molecule level. The authors reveal that anterograde IFT trains contain 40-50 dyneins while retrograde trains contain only 20-30, that the anterograde and retrograde dynein flux is similar (implying that there is no loss of dynein at the ciliary tip), that anterograde and retrograde IFT trains both contain anterograde and retrograde transport motors, and that the ratio of IFT dynein to IFT kinesin is constant along the cilium. Finally, the authors show for the first time that individual dynein motors can switch direction along the entire length of the cilium, in contrast to the unidirectional, dynein-driven transport of retrograde IFT trains. The manuscript is written with care and I feel that it will be a timely and well-cited contribution and thus recommend its publication by Nature Communications. However, before acceptance, the authors need to address the following minor comments:

1. The authors report that anterograde IFT trains contain 20-30 more dyneins than retrograde trains and that anterograde trains are longer than retrograde trains. However, the authors fail to discuss whether the length difference explains the difference in the number of bound dynein motors (assuming that the number of binding sites increases with the length of the train). In addition, the authors should analyze and discuss whether a higher number of shorter retrograde trains (compared to the number of the longer anterograde trains) could explain why the anterograde and retrograde flux of dynein is similar despite the different numbers of dynein motors bound to anterograde and retrograde trains.

2. I recommend that the authors compare the measured ensemble and single-molecule velocities with previously published velocities of retrograde IFT trains (see, e.g. Shih et al. 2013 eLife).

3. Finally, the authors should provide the attained single-molecule tracking precision.

Reviewer #3 (Remarks to the Author):

The study by Mijalkovic et al. is a very nice application of advanced imaging at the single molecule level inside a living organism. The authors study the dynamics of cytoplasmic dynein 2, the retrograde IFT motor within cilia. The study uses careful quantification of fluorescent ensemble and single molecule imaging, combined with modeling using the experimentally derived parameters, to conclude that the dynamics of motors binding to an IFT scaffolding, as well as the pausing and turnaround probability (which are likely a consequence of motor association/dissociation) are critical parameters controlling net motion and motor distribution along the cilia. The authors also find that motor numbers are relatively constant during transport, suggesting that directionality is not a consequence of motor ratio. The study and data presented appear to be of very high quality and will be of interest to the cilia and cytoskeleton communities. I support publication after minor revisions:

One big limitation of this and other studies on dynein-2 in cilia is the exclusive use of the light-intermediate chain as a reporter for dynein-2 in vivo. Given the findings of Li et al. 2015 (and Asante et al. 2014) that the diverse subunits of dynein-2 appear to undergo different dynamics within the cilia, it raises the question of what the authors are actually examining by imaging XBX-1? Is this a faithful reporter of the actual dynein-2 heavy chain, which the authors are most interested in observing? It would be useful for the field to have some data on the actual dynamics of the dynein-2 heavy chain, and this group appears to be in a good position to try that out given their expertise in genome editing and imaging. Given the big leap in this paper is the ability to image dynamics at the single molecule level, it would be all the more interesting if the authors could demonstrate that dynamics at this level are, or are not, specific to the subunit being imaged. For instance, it is not clear why in Li et al. the dynamics of XBX-1 do not match the dynamics of the intermediate chain, suggesting that the dynamics of the dynein-2 complex subunits are more complicated than appreciated, and they may not be stably assembled at any given time within the cilium. Since the conclusions of this paper rely on the assumption that XBX-1 reports on the motor (heavy chain) subunit itself, why not try and image that subunit directly? If the authors cannot address this point experimentally, I feel it warrants at least a discussion in the paper.

Further, I am confused why the author's could not image the distal tip of the cilia in this study but that region is imaged in Li et al, 2015. This becomes a critical point as Li et al. find that most XBX-1 particles turn around in this region, whereas the current manuscript cannot observe these events. This plays out in their inability to fully recapitulate their data in their mathematical model.

Minor comments:

1. Movie S2 is rich in information but plays relatively fast, necessitating multiple viewings. I suggest slowing the movie down by a factor of 2-3x.
2. I found the title of the manuscript a bit over the top.

Reviewer: Richard J. McKenney

Point-to-point response to reviewer comments for manuscript: "Backward, forward, pausing, turning: single-molecule behaviour required to maintain IFT dynein distribution in *C. elegans* chemosensory cilia"

Reviewer #1 (Remarks to the Author):

This study uses state-of-the-art in vivo imaging to study intraflagellar transport in *C. elegans*, in particular the behavior of fluorescent protein-tagged XBX-1, a light-intermediate chain of the retrograde IFT motor IFT dynein. IFT dynein is studied at the ensemble level (= IFT train) and the single molecule level. The manuscript shows high quality imaging (imaging single dynein motors in whole animals !!), is clearly written, and the data are solid.

We appreciate the referee's recognition of the technical advances in our manuscript that have made it possible to perform high-quality, quantitative imaging of single dynein motors in a living animal.

The manuscript addresses two main questions: First, considering that IFT trains are associated with both kinesins and dyneins, how is a tug-of-war avoided and processive transport of the ensembles over a distance of several microns ensured? The authors show that the ratio of anterograde and retrograde motors on the IFT trains is essentially constant. This is novel and interesting, but fits what was widely assumed and barely touches the question of how a tug-of-war between the opposing motors is avoided. The second question concerns the behavior of individual dynein motors and how it relates to and leads to the behavior of the ensembles (trains). Using modeling it is shown that the surprising variation of single molecule behavior (including diffusion, stationary, and turn-around) can explain the overall distribution of dynein. I have several questions about that approach and definiteness of the outcome (see below).

I feel that the current paper, while excellently executed and revealing various novel aspects of IFT, which will be of high interest to the cilia community, does not provide enough progress for publication in the journal.

Although this reviewer commends the quality and execution of our work, he or she is less enthusiastic than the other two reviewers about the potential impact. We respect this opinion, but do not agree. Our manuscript describes the study of dynein dynamics and motility at the single-molecule level in a living organism. The resulting biological insights are not only valuable for our understanding of IFT dynein in vivo, but also the complex and diverse functioning of other dynein motors. In addition, the technological advances and quantitative approach presented here are applicable to studies of molecular motors in general. Our findings highlight that the properties of a motor protein need to be considered in the context of the complete transport system: we show that IFT dynein shuttles between intervals in which it is active as a minus-end directed motor, transporting trains consisting of cargo together with other IFT dyneins and intervals in which it is a cargo, being transported by plus-end directed trains driven by kinesins. This results in complex behavior, including turnarounds and run lengths that are not only determined by the motor's interaction with the track (as would be measured in vitro) but also by the interactions with the rest of the train. Taking all this together, we are confident that our manuscript is of interest to the broad readership of a journal like Nature Communications.

Major points

Simulations were used to connect single molecule behavior to the overall distribution of IFT. I have

several questions about this approach.

1. I don't understand how the probabilities for anterograde-to-retrograde (p_{AR}) and retrograde-antegrade (p_{RA}) turns were obtained. Out of the 38 turnarounds observed just 4 were of the RA type. Retrograde IFT trains are 1.6 times more frequent than anterograde ones and that both AR and RA trains travel the same distance. The AR probability was given as 0.14/ μm and p_{RA} as 0.07/ μm . Considering that RA turnarounds are apparently much less frequent than AR turns (factor 8.5), how do the ps end up so similar (factor 2)?

As indicated in the manuscript, probability densities p_{AR} (A-to-R turnaround) and p_{RA} (R-to-A turnaround) – not probabilities but probabilities per micrometer traveled - were calculated using the total number of A-to-R (or R-to-A) turnarounds divided by the total anterograde (or retrograde) distance traveled (by all motors detected, also the ones not turning). In total, we observed and analyzed 135 retrograde and 182 anterograde moving XBX-1 (Figure 3A). Note also that the total distance traveled by the retrograde motors was 3.5 times less than the anterograde motors. The probability density we calculate is expressed per distance, not per time. This means that velocity also needs to be considered to connect these probability densities to the observed (relative) amount of turnarounds. As shown in Figure 1B and C in the manuscript, retrograde velocity is substantially larger than anterograde, and thus, in a given time window, less motors are moving retrograde than anterograde, resulting in substantially less than two times fewer R-to-A than A-to-R turnaround events.

2. The simulation assumes that dynein at the cilia tip turns-around without a pause. I wonder if that is a reasonable assumption considering data in other species documenting a pause of several seconds for IFT proteins at the cilia tip (see recent analysis by Reck et al. on the light intermediate chain D1bLIC-GFP and Buisson et al. in Trypanosoma; Reck et al. also determined A:R ratios in IFT frequency etc. and should be cited and discussed.)

In the new version of the manuscript we explicitly cite and discuss the relative A:R frequencies in Chlamydomonas (Reck et al.) and Trypanosoma (Buisson et al.), as suggested by the referee. We also compare dynein pausing times at the tip observed in Chlamydomonas with those in C. elegans (Reck et al.).

Indeed, in Chlamydomonas substantial pauses of ~1.7s have been observed at the tip for IFT dynein. Pausing has also been observed in Trypanosomes (Buisson 2013), but here the authors looked at an IFT particle and not the dynein motor. Our ensemble results (Figure 1) and the data of Li et al. suggest that dynein turns efficiently and quickly at the tip of C. elegans chemosensory cilia. In our single-molecule data we could not observe IFT dynein close to the tip, since the phasmid cilia are slightly bent. In all our original data the tip of the cilium is therefore not perfectly in focus, while the rest of the cilium is. This made it impossible for us to measure pauses at the tip. In a new set of experiments, we have now imaged the small fraction of worms that is oriented such that the tip is in the same focal plane as most of the cilium. We have added this new data to the manuscript, focusing on the tips of these worms. The data shows that most XBX-1 reverse direction in less than 600ms (Figure S4). These new experimental findings justify the assumption made in our simulations and are in agreement with the ensemble data in C. elegans.

3. The conclusions are based on the analysis of a single marker (XBX-1). Playing devil's advocate, I wonder to which extent XBX-1's behavior is representative for the entire IFT dynein complexes? Or that the GFP-tag results in an aberrant behavior of the protein/protein complex?

We recognize the concern raised here by the reviewer. We cannot completely exclude that XBX-1 could behave differently than other parts of dynein complex (and have now addressed this in the Discussion section of the manuscript).

*We note here, however, that (i) XBX-1 is a well-established component of the IFT dynein complex in *C. elegans* and other organisms (human (D2LIC; Grissom et al 2001) and *Chlamydomonas* (LIC, Perrone et al 2003) orthologues were shown to co-precipitate with the dynein-2 heavy chain), (ii) that XBX-1 is a component essential for the IFT dynein complex to perform retrograde transport (Schafer et al 2003) and (iii) that it has been used frequently in the literature as a marker of IFT dynein, both with a YFP (Schafer et al 2003; Hao et al 2011) and GFP tag (Burghoorn et al 2010, Wei et al 2012).*

It is difficult to completely exclude that the GFP-tag has an effect on the labeled protein. What we do know is that velocities and other IFT parameters (train frequency) are very similar to what we (and others) have observed before using markers on other IFT components (kinesins, particles).

It should be mentioned somewhere that this light chain is stably integrated into the dynein complex (co-purifies?? reference?).

Indeed, we acknowledge that our choice for XBX-1 labeling could have been documented and discussed better in the previous version of the manuscript. We have now changed this and added additional explanations and references in this regard. In addition, we have more carefully written in our manuscript that our results are for XBX-1.

More difficult to exclude is the possibility that entire pieces of an IFT train (dynein + IFTA and/or IFT-B) brake-off and re-attach. Then, the observed single molecule features would be those of the IFT train subunits rather than of the dynein motor. This could actually be tested experimentally using an IFT particle proteins and/or a second IFT dynein subunit. Or do the authors dismiss this possibility? *The reviewer is correct that detachment of part of a train could, in principle, also explain our results. We have dismissed this possibility because in a previous study (Prevo et al. 2015) we looked (among others) at the behavior of IFT-B. We did not observe turnaround of IFT-B in the proximal and distal segment, only at the tip. We thus favor the explanation that individual dynein motors fall off trains (just like we have seen in the Prevo study for the kinesins) and can be picked up by another train moving in the opposite direction. We have modified the Discussion to address this more clearly in the manuscript.*

Minor points

The terms 'backward', 'switch direction' and similar could be easily misinterpreted as dynein actually taking a step as a motor toward the microtubule plus-end as it has been reported for dynein-1. Actually, XBX-1 switches from being a cargo (A) to being an active (or inactive) motor (R).

The text in the Results section has been modified to clarify the use of term “switch direction”. In the previous version of the manuscript we had chosen not to use the terms “forward” and “backward” in the main text, but used “anterograde” and “retrograde” instead. We now have also removed “forward” and “backward” from the title.

The axes in the plot in Fig. 3b should be oriented as in the other subpanels (c-f). Also, it would be nice to show kymographs in addition to the particle tracings.

The orientation of the plot in Fig 3B was changed to maintain consistency with the other plots in Figure 3. We have also added kymographs corresponding to Figure 3E and 3F.

p.4 length of A and R trains: reference 28 has been superseded by Vannuccini et al. 2016 and especially Stepanek and Gaia 2016.

These references have been added and discussed.

p. 12, "...roles we cannot distinguish between." However, in several other places in the manuscript (e.g., line 308/309 on the same page) the authors do distinguish and assume that IFT dynein is a cargo on anterograde trains (reasonable) but active in retrograde trains (probable, but who knows how many of the dyneins in such an ensemble are actively moving and how many are carried along as cargo? ***The referee is right: we can be quite sure that the dyneins on an anterograde train are not active and can be considered cargo. On the other hand, on a retrograde moving train, in principle, all dynein motors could be driving the transport, but there is currently no way of deciphering which of the motors is actually engaged. Some might indeed not be active and could be considered cargo. We have explained this point in the Discussion.***

Reviewer #2 (Remarks to the Author):

The manuscript by Mijalkovic et al. reports on the successful (genetic) fluorescence tagging of cytoplasmic dynein 2 (IFT dynein) in *C. elegans* strains and the imaging of the tagged dyneins in vivo both at the ensemble and single-molecule level. The authors reveal that anterograde IFT trains contain 40-50 dyneins while retrograde trains contain only 20-30, that the anterograde and retrograde dynein flux is similar (implying that there is no loss of dynein at the ciliary tip), that anterograde and retrograde IFT trains both contain anterograde and retrograde transport motors, and that the ratio of IFT dynein to IFT kinesin is constant along the cilium. Finally, the authors show for the first time that individual dynein motors can switch direction along the entire length of the cilium, in contrast to the unidirectional, dynein-driven transport of retrograde IFT trains. The manuscript is written with care and I feel that it will be a timely and well-cited contribution and thus recommend its publication by Nature Communications. However, before acceptance, the authors need to address the following minor comments:

We appreciate the enthusiastic response of this referee to our work and have responded by making the requested changes.

1. The authors report that anterograde IFT trains contain 20-30 more dyneins than retrograde trains and that anterograde trains are longer than retrograde trains. However, the authors fail to discuss whether the length difference explains the difference in the number of bound dynein motors (assuming that the number of binding sites increases with the length of the train).

Our finding that retrograde trains contain less dyneins than anterograde corroborates well with our earlier observation (Prevo et al. 2015) for IFT-A, IFT-B markers and both kinesins. Retrograde trains contain about half the amount of IFT-A and IFT-B than anterograde. So indeed, number of dyneins appears correlated with train size. We now address this in the manuscript (Results section).

In addition, the authors should analyze and discuss whether a higher number of shorter retrograde trains (compared to the number of the longer anterograde trains) could explain why the anterograde and retrograde flux of dynein is similar despite the different numbers of dynein motors bound to anterograde and retrograde trains.

This is indeed exactly what we find: compared to anterograde trains, retrograde trains are smaller but more frequent. Smaller we infer from the train fluorescence intensities, more frequent from the higher amount of kymograph lines observed (figure 1F). This latter determination should be seen as a strong indication; it might, however, (as also discussed before by Buisson et al. 2013) not be very accurate, since very small trains might not be resolvable in the kymographs. Our flux determination (figure 1G) takes into account all motors, also those in small trains. We have revised the text to avoid confusion.

2. I recommend that the authors compare the measured ensemble and single-molecule velocities with previously published velocities of retrograde IFT trains (see, e.g. Shih et al. 2013 eLife).

We have modified the manuscript to include a comparison with previously published velocities of retrograde IFT trains, including Shih et al 2013.

3. Finally, the authors should provide the attained single-molecule tracking precision.

We have added the single-molecule tracking precision to the Materials and Methods section of the manuscript.

Reviewer #3 (Remarks to the Author):

The study by Mijalkovic et al. is a very nice application of advanced imaging at the single molecule level inside a living organism. The authors study the dynamics of cytoplasmic dynein 2, the retrograde IFT motor within cilia. The study uses careful quantification of fluorescent ensemble and single molecule imaging, combined with modeling using the experimentally derived parameters, to conclude that the dynamics of motors binding to an IFT scaffolding, as well as the pausing and turnaround probability (which are likely a consequence of motor association/dissociation) are critical parameters controlling net motion and motor distribution along the cilia. The authors also find that motor numbers are relatively constant during transport, suggesting that directionality is not a consequence of motor ratio. The study and data presented appear to be of very high quality and will be of interest to the cilia and cytoskeleton communities. I support publication after minor revisions:

We very much appreciate the enthusiastic response of this referee to our work. The minor revisions suggested by the reviewer have been addressed in the Results and Discussion sections of the manuscript.

One big limitation of this and other studies on dynein-2 in cilia is the exclusive use of the light-intermediate chain as a reporter for dynein-2 in vivo. Given the findings of Li et al. 2015 (and Asante et al. 2014) that the diverse subunits of dynein-2 appear to undergo different dynamics within the cilia, it raises the question of what the authors are actually examining by imaging XBX-1? Is this a faithful reporter of the actual dynein-2 heavy chain, which the authors are most interested in observing? ***Evidence in literature points towards XBX-1 being a crucial part of the C. elegans dynein-2 motor. In C. elegans, mutations in XBX-1 disrupt retrograde IFT and there is no retrograde transport of***

XBX-1 in the che-3 (heavy chain) mutant (Schafer et al 2003). Additionally, both its human (D2LIC; Grissom et al 2001) and Chlamydomonas (LIC, Perrone et al 2003) orthologues were shown to co-precipitate with the dynein-2 heavy chain. Together, these findings provide a strong body of evidence that XBX-1 is an essential component of IFT dynein.

The dynamics of other subunits associated with dynein-2, as investigated by Asante et al 2014 and Li et al 2015, remains an intriguing research question. One of the challenges in the field is that the exact composition and structure of dynein-2 subunits remains unknown. Neither of the studies mentioned (including our own) reports on the dynamics of the IFT dynein heavy chain, which includes the ATP and microtubule binding sites and would thus be the most ideal candidate for labeling (but see below). Any other components, including those investigated by Li et al 2015 (DLC-1, DYCI-1, DYLT-3 and XBX-1), could in theory also be adapter, regulatory or cargo molecules. It is our opinion that, out of these components, the evidence in literature is most strong that XBX-1 is an integral part of the motor, as reflected by the number of studies that have used this subunit as a dynein-2 reporter. But the reviewer is right, at this point we cannot exclude that the composition of the IFT dynein complex is dynamic. We have added a discussion of this issue to the manuscript and made it more clear that we have direct evidence for IFT-dynein subunit XBX-1 alone.

It would be useful for the field to have some data on the actual dynamics of the dynein-2 heavy chain, and this group appears to be in a good position to try that out given their expertise in genome editing and imaging. Given the big leap in this paper is the ability to image dynamics at the single molecule level, it would be all the more interesting if the authors could demonstrate that dynamics at this level are, or are not, specific to the subunit being imaged.

For instance, it is not clear why in Li et al. the dynamics of XBX-1 do not match the dynamics of the intermediate chain, suggesting that the dynamics of the dynein-2 complex subunits are more complicated than appreciated, and they may not be stably assembled at any given time within the cilium. Since the conclusions of this paper rely on the assumption that XBX-1 reports on the motor (heavy chain) subunit itself, why not try and image that subunit directly? If the authors cannot address this point experimentally, I feel it warrants at least a discussion in the paper.

We acknowledge that investigating the single-molecule dynamics of various possible dynein-2 components would be valuable. For our current manuscript, this is, however, not feasible within a reasonable time frame. When we generated the C. elegans strains for our manuscript, MosSCI was the state-of-the-art approach to create recombinant strains. MosSCI has its limitations with respect to the maximum length of DNA that can be inserted and Che-3 (the gene coding for the heavy chain) far exceeds what is possible. Nowadays, CRISPR is the state-of-the-art. And this approach might be able to do the job, although, for example, Li et al. who address more or less the point raised by the reviewer have not shown data on Che-3::EGFP, only on the smaller components. We appreciate the points raised by the reviewer regarding the complexity of dynein-2 component dynamics and have included these points of Discussion in the manuscript.

Further, I am confused why the author's could not image the distal tip of the cilia in this study but that region is imaged in Li et al, 2015. This becomes a critical point as Li et al. find that most XBX-1 particles turn around in this region, whereas the current manuscript cannot observe these events. This plays out in their inability to fully recapitulate their data in their mathematical model.

The phasmid cilia of C. elegans are not straight; the last 1-2 micrometers of the distal segments

bend outward. Using confocal microscopy of TBB-4::EGFP (tubulin) we have seen that in ~95% of the cases (i.e. worms) the tip of the cilium does not lie in the same focal plane as the remainder of the cilium (unpublished data). In single-molecule imaging no molecules can be observed when they are a couple of 100s of nanometers out of focus. In higher intensity train studies (like our Fig. 1 and Li et al.) the out-of-focus tip can still be observed, but is blurred. In our quantitative analyses we do not take this out-of-focus stretch into account. Qualitative inspection of our and Li's train kymographs reveals that XBX-1 trains efficiently turn at the tip, without a substantial delay. We have clarified this in the text.

As the referee points out, the simulation in Figure 4 relies on the assumption that XBX-1 reverses direction without delay at the tip. We now present new single-XBX-1 data (of the ~5% of worms in an orientation in which the distal part of the cilium, including tip, is in focus) that shows that turns occur at the tip almost instantaneously (in under 600ms; Figure S4), justifying the assumptions we used for our modeling.

Minor comments:

1. Movie S2 is rich in information but plays relatively fast, necessitating multiple viewings. I suggest slowing the movie down by a factor of 2-3x.

We have slowed down Movie S2 (by a factor of 2) as suggested.

2. I found the title of the manuscript a bit over the top.

We changed the title to: "Ensemble and single-molecule dynamics of IFT dynein in Caenorhabditis elegans cilia"

REVIEWERS' COMMENTS:

Reviewer #1 (Remarks to the Author):

Mijalkovic et al.

In their revision, the authors have addressed most of my concerns and improved the manuscript, e.g., by adding direct data on the turnaround of XBX1 at the tip. The data support a model in which IFT trains move processively along the cilia whereas the IFT motors exit and rejoin the moving IFT trains. The work provides a significant step ahead in our understanding of IFT traffic. As so often, in vivo imaging reveal a much more complex behavior of proteins than previously anticipated. The authors nicely demonstrate that this behavior of single molecules corresponds/results/causes to the overall distribution of xbx1 inside cilia. My concern was and is that the importance of this on-and-off behavior of the motors for IFT remains elusive and that the manuscript in no way tries to address that question experimentally, e.g., by comparing single molecule behavior of xbx1 and its overall distribution in *osm3* or *bbs* mutants. Thus the manuscript is descriptive but description and observation are the first step to understanding a phenomenon. I write this solely to challenge the authors to dig deeper and not to depreciate this nice study.

Other comments

"The motors, kinesin-II 10, OSM-3 10 and IFT dynein (this work), gradually dock on and off to this scaffold in a tightly controlled way, with conserved flux. Tight control of the binding and release of the motors ensures that the motors are distributed differently along the cilia: the importer kinesin-II enriched at base and transition zone, the long-distance transporter OSM-3 in proximal and distal segment and the sole retrograde motor IFT dynein relatively constant along the cilium."

This conclusion should probably be limited to *C. elegans* since many organisms build cilia of considerable length without employing the "long distance" kinesin but depend solely on the "importer" kinesin-2 for cilia assembly. This may also serve as an example how over-interpretation of imaging data can lead to short-lived models. The recent observation that kinesin-2 moves along the B-tubules whereas dynein1b moves along the A-tubules (Pigino lab) could explain the use of two motors in *C. elegans* with its long singlet MT distal segment just as well or even better; it also fits with the phenotype of *osm3* mutants.

"In contrast, in *C. reinhardtii*, equal anterograde and retrograde train frequencies have been described.[ref36]"

This statement is incorrect and based on a rescue strain not wild type; see earlier figure in the same paper demonstrating a clear difference in the frequency of a and r transports. *C. reinhardtii* is misspelled repeatedly in the manuscript.

"As such, these values do not correspond to run lengths 329 measured in vitro on isolated motors using optical tweezers or single-molecule fluorescence 330 microscopy, which have not been determined for IFT dynein, but are typically 1-3 μm for 331 cytoplasmic dynein."

But see Shih et al. *elife* 2013 for estimates on the number of active dyneins in retrograde trains

Reviewer #2 (Remarks to the Author):

The revision has significantly strengthened the manuscript and satisfactorily answered my comments. I recommend publication.

Reviewer #3 (Remarks to the Author):

The authors have done a satisfactory job of responding to my comments. I support publication of

the manuscript as is with the following minor revision:

A final point was raised by the inclusion of the kymographs of single dynein behavior in Fig. 3E,F. These kymographs are small and should be made bigger to allow the reader to discern features for themselves. In Fig. 3F, it is hard for me to tell because of the size of the picture, but the kymograph and diagram drawn next to it don't look like they have the same shape trajectory to me. Also importantly, does the fluorescence intensity of the particle that turns around change? It appears so to me in this small image. In addition to larger kymographs, perhaps the inclusion of these examples as a supplementary movie would help the reader as well.

Point-by-point response to reviewer comments:

Reviewer #1 (Remarks to the Author):

In their revision, the authors have addressed most of my concerns and improved the manuscript, e.g., by adding direct data on the turnaround of XBX1 at the tip. The data support a model in which IFT trains move processively along the cilia whereas the IFT motors exit and rejoin the moving IFT trains. The work provides a significant step ahead in our understanding of IFT traffic. As so often, in vivo imaging reveal a much more complex behavior of proteins than previously anticipated. The authors nicely demonstrate that this behavior of single molecules corresponds/results/causes to the overall distribution of *xbx1* inside cilia. My concern was and is that the importance of this on-and-off behavior of the motors for IFT remains elusive and that the manuscript in no way tries to address that question experimentally, e.g., by comparing single molecule behavior of *xbx1* and its overall distribution in *osm3* or *bbs* mutants. Thus the manuscript is descriptive but description and observation are the first step to understanding a phenomenon. I write this solely to challenge the authors to dig deeper and not to deprecate this nice study.

Other comments

“The motors, kinesin-II 10, OSM-3 10 and IFT dynein (this work), gradually dock on and off to this scaffold in a tightly controlled way, with conserved flux. Tight control of the binding and release of the motors ensures that the motors are distributed differently along the cilia: the importer kinesin-II enriched at base and transition zone, the long-distance transporter OSM-3 in proximal and distal segment and the sole retrograde motor IFT dynein relatively constant along the cilium.” This conclusion should probably be limited to *C. elegans* since many organisms build cilia of considerable length without employing the “long distance” kinesin but depend solely on the “importer” kinesin-2 for cilia assembly. This may also serve as an example how over-interpretation of imaging data can lead to short-lived models. The recent observation that kinesin-2 moves along the B-tubules whereas dynein1b moves along the A-tubules (Pigino lab) could explain the use of two motors in *C. elegans* with its long singlet MT distal segment just as well or even better; it also fits with the phenotype of *osm3* mutants.

We have now more clearly indicated in the manuscript that these conclusions apply primarily to IFT in C. elegans. Little is known about kinesin-2 co-operation in other organisms and the dynamics of IFT motors could indeed be different there.

“In contrast, in *C. reinhardtii*, equal anterograde and retrograde train frequencies have been described.[ref36]”

This statement is incorrect and based on a rescue strain not wild type; see earlier figure in the same paper demonstrating a clear difference in the frequency of a and r transports.

We have corrected this in the manuscript.

C. reinhardtii is misspelled repeatedly in the manuscript.

We have corrected this in the manuscript.

“As such, these values do not correspond to run lengths measured in vitro on isolated motors using optical tweezers or single-molecule fluorescence microscopy, which have not been determined for IFT dynein, but are typically 1-3 μm for cytoplasmic dynein.”

But see Shih et al. *elife* 2013 for estimates on the number of active dyneins in retrograde trains.

We have included the Shih et al reference as follows: “From in vitro motor stall force measurements it has been estimated that at least four motors actively drive transport at a given time in C. reinhardtii³⁰, but the rest might effectively be cargo. Here we cannot distinguish between these two roles.”

Reviewer #2 (Remarks to the Author):

The revision has significantly strengthened the manuscript and satisfactorily answered my comments. I recommend publication.

Reviewer #3 (Remarks to the Author):

The authors have done a satisfactory job of responding to my comments. I support publication of the manuscript as is with the following minor revision:

A final point was raised by the inclusion of the kymographs of single dynein behavior in Fig. 3E,F. These kymographs are small and should be made bigger to allow the reader to discern features for themselves. In Fig. 3F, it is hard for me to tell because of the size of the picture, but the kymograph and diagram drawn next to it don't look like they have the same shape trajectory to me. Also importantly, does the fluorescence intensity of the particle that turns around change? It appears so to me in this small image. In addition to larger kymographs, perhaps the inclusion of these examples as a supplementary movie would help the reader as well.

The kymographs have been enlarged, as suggested by the reviewer. The fluorescence intensity of turning particles does not change, as expected for a single molecule. Trajectories are only included in the analysis, after inspection of the fluorescence intensity has confirmed that they are indeed due to a single molecule. The trajectory from Figure 3F was already included in Supplementary Movie 2 as an example of an A-to-R turn. We have now added an additional reference in the text to this Supplementary Movie, as well as adapted the legend of the Supplementary Movie and Figure 3.